# Exfoliating effect of β-glycyrrhetinic acid on plaque inducing gingivitis: Comparison with cetylpyridinium chloride

Shinya Kato[1,2,3], Xiangtao Ma[3,4], Kayo Sato[5], Aya Okumura[5], Nobuo Yoshinari[1,3], Akihiro Yoshida [2,3]*

1 Department of Periodontology, Faculty of Dentistry, Matsumoto Dental University, Shiojiri, Japan,
2 Department of Oral Microbiology, Faculty of Dentistry, Matsumoto Dental University, Shiojiri, Japan,
3 Department of Oral Health Promotion, Graduate School of Oral Medicine, Matsumoto Dental University, Shiojiri, Japan, 4 Hospital of Stomatology, Hebei Medical University, Hebei, China, 5 Human Health Care Product Research, Kao Corporation, Tokyo, Japan

* akihiro.yoshida@mdu.ac.jp

## Abstract

β-glycyrrhetinic acid (BGA) possesses antibacterial effects against human supragingival plaque bacteria and inhibits biofilm formation. However, the effect of BGA on already formed dental plaque has not been investigated. We analyzed and compared the effects of BGA on preformed supragingival plaque biofilms with those of cetylpyridinium chloride (CPC). All experiments were performed *in vitro* using biofilms formed by incubating supragingival plaque bacteria for 24 h. First, we analyzed the number of viable and dead bacteria in the biofilms following BGA and CPC application. The number of viable bacteria was significantly reduced by BGA treatment compared with by the solvent control. However, the viable/dead bacterial ratios did not significantly vary. Conversely, the turbidity in the supernatants (optical density at 600 nm [$OD_{600}$]) was $0.237 \pm 0.003$, $0.136 \pm 0.002$, and $0.096 \pm 0.002$ for the BGA, CPC, and solvent control groups, respectively, indicating superior ability of BGA in biofilm exfoliation. This study suggests that BGA may inhibit dental plaque adhesion and eliminate and disinfect dental plaque through a mechanism different from CPC, contributing to the prevention of periodontal disease.

## Introduction

Stability of the host-microbial interface across mucosal surfaces in the human body is essential for the maintenance of oral health [1]. This is especially relevant concerning the mucosal surfaces, which present a constant microbial challenge in the host epithelial barriers [2]. Under oral conditions, commensal bacteria actively interact with the gingival tissue to maintain healthy neutrophil surveillance and normal tissue and bone turnover processes [3–5]. The disruption of this homeostatic host-bacteria relationship occurs during plaque-induced gingivitis, marking the initiation of

**Data availability statement:** All relevant data are within the paper and its Supporting Information files.

**Funding:** This research was funded by the Kao Corporation.

**Competing interests:** I have read the journal's policy and the authors of this manuscript have the following competing interests: Kayo Sato and Aya Okumura are paid employees of Kao Corporation, Tokyo, Japan. This research was funded by Kao Corporation. The authors declare no conflicts of interest.

periodontitis [6–8]. Plaque-induced gingivitis is caused by substances derived from microbial plaque accumulation at or near the marginal gingiva; thus, an increase in the bacterial burden increases gingival inflammation [9–13]. Localized inflammation caused by dental plaque can lead to oxygen deprivation in the area and promote the growth of anaerobic microorganisms, triggering the onset of periodontitis [14]. When gingivitis progresses to periodontitis, the alveolar bone is pathologically resorbed, causing tooth loss [15]. Therefore, controlling plaque-induced gingivitis is important for controlling periodontal disease for tooth preservation.

Cetylpyridinium chloride (CPC) is a cationic quaternary ammonium compound with surface-active properties. Its mechanism of action relies on the hydrophilic part of the CPC molecule interacting with the bacterial cell membrane leading to loss of cell components, disruption of cell metabolism, inhibition of cell growth, and finally cell death. It has a broad antimicrobial spectrum, with rapid killing of gram-positive pathogens and yeast in particular [16]. CPC exhibits anti-plaque and anti-gingivitis effects [17]. However, the use of CPC reportedly results in the growth of resistant bacteria. Although no studies have demonstrated the long-term effects of CPC exposure on the oral flora, concerns about the emergence of CPC-resistant oral bacteria exist [18]. Moreover, antimicrobial agents, such as CPC, influence the flora of other organs, such as the intestinal tract, and emergence of drug-resistant bacteria [19].

Therefore, conventional antimicrobial agents should not be used extensively in the treatment of gingivitis and periodontitis. New anti-infection strategies are needed to control the spread of resistant bacteria. Biofilm control warrants the use of antimicrobial substances that do not rely on traditional antimicrobials owing to their low sensitivity to bacteria in biofilms and their potential to increase the number of resistant bacteria [20,21].

*Glycyrrhiza glabra*, also known as licorice, is a herbaceous perennial plant that has been used as a therapeutic agent for thousands of years. β-glycyrrhetinic acid (BGA) is obtained by hydrolyzing glycyrrhizic acid extracted from licorice [22,23] and has been reported to have strong anti-inflammatory [24–26], antioxidative [27], and antibacterial activities [28–31]. Previous investigations confirmed that BGA reduced the biofilm formation and virulence expression of *Pseudomonas aeruginosa,* a representative multidrug-resistant species [28,32]. Another study reported that BGA promoted cell survival and reduced pro-inflammatory cytokines production, during carbapenem-resistant Klebsiella pneumoniae-induced human pulmonary epithelial cell injury [33]. Therefore, BGA is effective against drug-resistant bacteria, including multidrug-resistant bacteria, which are increasingly emerging owing to inappropriate use of antimicrobial agents [30,34]. BGA-containing dentifrices reportedly inhibited the accumulation of supragingival plaque in a clinical study [35].

We previously elucidated the inhibitory effect of BGA on supragingival plaque formation [36]. However, the effect of BGA on already formed dental plaque has not been investigated. The present study aimed to analyze the effect of BGA on previously formed supragingival plaque and compare the effect of BGA on supragingival biofilm with that of CPC.

## Materials and methods

### Ethical considerations

All procedures were conducted in accordance with the guidelines of the Ethics Committee of the Faculty of Dentistry, Matsumoto Dental University (No. 0295) and the Declaration of Helsinki (64th WMA General Assembly, Fortaleza, October 2013) [37]. For assays using human supragingival plaques, plaque samples were obtained from healthy volunteers after obtaining written informed consent.

### Bacterial strains and culture

The bacterial strains used in this study are listed in Table 1. All the bacteria were cultured as described previously [36]. Briefly, *Streptococcus* and *Actinomyces* species were inoculated in Bacto™ Brain Heart Infusion (BHI) (BD Biosciences, Franklin Lakes, NJ, USA) broth at 37 °C under anaerobic conditions [38]. *Aggregatibacter actinomycetemcomitans* was inoculated in trypticase soy broth (Becton Dickinson, Sparks, MD, USA) supplemented with 0.6% yeast extract (Becton Biosciences) and 0.04% sodium bicarbonate at 37 °C in a 5% $CO_2$ atmosphere. *Prevotella* spp., *Fusobacterium nucleatum*, and *Porphyromonas gingivalis* were grown in Gifu Anaerobic Medium (GAM; Nissui Medical Co., Tokyo, Japan) at 37

**Table 1. Bacterial strains and MICs and MBCs of BGA.**

| Bacterial species | Strains | MIC (µg/mL) | MBC (µg/mL) | MBC/MIC |
|---|---|---|---|---|
| *Streptococcus mutans* | MT 8148 | 128 | 1024 | 8 |
| | Ingbrid | 32 | 1024 | 32 |
| | XC | 32 | 1024 | 32 |
| | OMZ175 | 32 | 1024 | 32 |
| | 10449 | 32 | 1024 | 32 |
| *Streptococcus sobrinus* | GTC278 | 16 | 1024 | 64 |
| | 6715 | 32 | 512 | 16 |
| *Streptococcus anginosus* | NTCT10713 | 64 | 1024 | 16 |
| *Streptococcus mitis* | 9811 | 64 | 1024 | 16 |
| *Streptococcus sanguinis* | ATCC10556 | 64 | 1024 | 16 |
| *Streptococcus salivarius* | JCM5707 | 512 | 1024 | 2 |
| | HHT | 64 | 512 | 8 |
| | HT9A | 128 | 1024 | 8 |
| *Streptococcus gordonii* | DL1 | 64 | 512 | 8 |
| *Streptococcus oralis* | 557 | 64 | 512 | 8 |
| *Actinomyces naeslundii* | ATCC12104 | 64 | 512 | 8 |
| *Actinomyces viscosus* | ATCC15987 | 64 | 1024 | 16 |
| *Aggregatibacter actinomycetemcomitans* | JP2 | 128 | 512 | 4 |
| | Y4 | 64 | 1024 | 16 |
| *Prevotella denticola* | JCM8528 | 128 | 512 | 4 |
| *Prevotella nigrescens* | ATCC33563 | 64 | 2048 | 32 |
| *Fusobacterium nucleatum* | JCM6328 | 128 | 1024 | 8 |
| *Porphyromonas gingivalis* | W83 | 32 | 2048 | 64 |
| | ATCC33277 | 64 | 512 | 8 |
| *Prevotella intermedia* | ATCC25611 | 256 | 1024 | 4 |

MIC, minimum inhibitory concentrations; MBC, minimum bactericidal concentration; BGA, β-glycyrrhetinic acid.

°C under anaerobic conditions. *P. gingivalis* was inoculated in GAM broth supplemented with 5 µg of hemin per mL, 1.0 µg of menadione per mL, and 1.0% L-cysteine at 37 °C under anaerobic conditions [39].

## Supragingival plaque collection

Supragingival plaque samples were collected using a sterile curette from the mandibular left first molars of 12 healthy participants. To investigate the efficacy in preventing gingivitis and periodontitis, bacterial communities adjacent to the gingival margin were collected from participants who did not have gingivitis or periodontitis. The age of the participants ranged 26–58 years, with an average age of $36.0 \pm 9.8$ years. Plaque samples were collected immediately above the gingival margin into 1.0 mL of sterile phosphate-buffered saline (PBS, FUJIFILM Wako Pure Chemical Co. Osaka, Japan). Plaque samples collected from the 12 volunteers were combined in equal proportions to form single biofilms. Using this biofilm enabled consistent experimentation with a uniform biofilm, requiring minimal sample collection per individual. This mixture of supragingival plaque was stored at −20 °C until use and incubated in BHI medium (Becton Dickinson and Co.) to a cell density of 1.0 optical density at 600 nm ($OD_{600}$), and then used as a supragingival plaque solution for experiments.

## BGA and cetylpyridinium chloride

BGA was obtained commercially (Alps Pharmaceutical, Inc., Co., Ltd., Gifu, Japan), and dissolved in 100% dimethyl sulfoxide (DMSO) (FUJIFILM Wako Pure Chemical Co.). Stock solutions of the reagents were prepared at a concentration of 128 mg/mL and diluted in the medium to the appropriate concentrations for each experiment.

CPC (FUJIFILM Wako Pure Chemical Co.) was used to compare with BGA.

## Minimum inhibitory concentrations and minimum bactericidal concentrations determination of BGA

The antibacterial activity of BGA was evaluated by determining the minimum inhibitory concentrations (MICs) and minimum bactericidal concentrations (MBCs) using the microdilution method, as previously described [40]. Briefly, BGA was adjusted to 1024 µg/mL in BHI and two-fold serial dilutions were prepared in 96-well microplates (0, 8, 16, 32, 64, 128, 256, 512, and 1024 µg/mL; 96-well culture plate U-shape bottom, WATSON, Tokyo, Japan). Overnight bacterial cultures were adjusted to an $OD_{600}$ of 1.0 ($10^8$ cells/mL) and diluted to 1:100 with 100 µL of BHI ($10^6$ cells/mL). To each well, 10 µL of bacterial cultures was added, resulting in 100-µL cultures. The MICs of BGA were determined after 24 h of anaerobic incubation at 37°C. From the wells where bacteria did not grow, the medium was incubated on GAM agar medium (Nissui Medical Co.) supplemented with 5 µg of hemin per mL, 1.0 µg of menadione per mL, and 1.0% L-cysteine without antimicrobials, and the BGA concentration with a bacterial count of 10 or less was used as the MBC. When the MBC was 1024 µg/ml or higher, the BGA concentration was adjusted to 0, 256, 512, 1024, 2048, and 4096 µg/ml, and the experiment was repeated using the same procedure.

## Analysis of the number of bacteria in the biofilms

Flat-bottomed polystyrene microtiter plates (96-well Easy Wash; Corning Inc., Corning, N.Y.) containing 100 µL of BHI per well were inoculated with 1 µL supragingival plaque solution obtained from 12 volunteers for 24 h at 37 °C to form biofilms [41]. The formed biofilms were washed with PBS and incubated in test media (i.e., the BHI medium with 128 µg/mL BGA and 40 µg/mL CPC dissolved in DMSO, and BHI medium with only DMSO without antibiotic) (S1 Table) for an additional 6 h. The colony-forming units (CFU) in the biofilm remaining at the bottom of the plate were measured using the 10-fold dilution method.

## LIVE/DEAD staining

To each well of an 8-well plate (Nunc Lab-Tek II Chamber Slide System, Thermo Fisher Scientific, Waltham, MA), 495 µL of BHI and 5 µL of supragingival plaque solution were added and incubated for 24 h to form biofilms [41]. After biofilm

formation, test media (S1 Table) were added to each well and allowed to incubate for 6 h. The LIVE/DEAD BacLight Bacterial Viability Kit (L7012, Invitrogen, Mount Waverley, Australia) was used for 15 min, and LIVE/DEAD staining was performed according to the manufacturer's instructions.

### Confocal laser scanning microscope analysis of biofilms

LIVE/DEAD-stained biofilms were imaged using confocal laser scanning microscopy (CLSM, Axiovert 200M Inverted Microscope, Carl Zeiss, Jena, Germany) and rendered in the x–y–z planes using ZEN 3.6 software (Carl Zeiss) for analyzing the bactericidal effect. In accordance with previous reports, the proportion of nonviable bacteria, based on the green (viable cells) and red (dead/damaged cells) pixel intensities for every pixel in the x–y–z planes, was evaluated using ImageJ software (National Institutes of Health [NIH]) [42].

### Analysis of the biofilm amount

Flat-bottomed polystyrene microtiter plates (96-well Easy Wash; Corning Inc.) containing 100 μL of BHI per well were inoculated with supragingival plaque solution and incubated for 24 h at 37 °C to form biofilms [41]. After biofilm formation, test media (S1 Table) were added to each well and allowed to stand for 6 h. After incubation, 25 μL of 1% (wt/vol) crystal violet (CV) solution was added to each well. After 15 min, the wells were rinsed three times with 200 μL of distilled water and air dried. The CV on the abiotic surfaces was solubilized in 95% ethanol and the $OD_{600}$ was measured [43].

### Analysis of the exfoliating action of BGA on biofilms

Biofilms were formed on 6-well polystyrene plates (Corning Inc.) using a supragingival plaque solution and incubated for 6 h following addition of the test media (S1 Table). The amount of suspended solids was analyzed by measuring the absorbance at $OD_{600}$ using a microplate reader (iMark microplate reader; Bio-Rad Laboratories Inc., Hercules, CA, USA). The CFUs in the supernatants were measured using the 10-fold dilution method.

### Analysis of the embrittlement effect of BGA on biofilms

Biofilms were formed in six-well plates, incubated for 6 h following the addition of the test media (S1 Table). The biofilm was then subjected to physical vibration (amplitude: 20%, 1 pulse) using an ultrasonic sonication machine (UP-200S; Hielscher Ultrasonics GmbH, Teltow, Germany), and the amount of exfoliated biofilm was quantified to analyze the effect of antimicrobial agents on deterioration of the biofilm. The amount of biofilm formed before and after physical vibration was determined using a CV assay [43].

### Statistical analysis

Statistical analyses were performed using SPSS Statistics version 28.0.0.0 (IBM Corp., Armonk, NY, USA). The amount of biofilm, the number of viable bacteria and the viable/dead bacterial ratios in the biofilms, and the turbidity and the number of viable bacteria in the supernatants after test solution application were analyzed using one-way analysis of variance for a priori comparisons and the Scheffé test for a post-hoc test to compare between the three groups (DMSO, CPC, and BGA). Comparisons before and after ultrasound were performed using a paired t-test. Normality was tested by Shapiro–Wilk test. A $p < 0.05$ was considered statistically significant.

## Results

### Bactericidal effects of BGA on oral bacteria

In this study, we analyzed the MBCs to determine the bactericidal effect of BGA (Table 1). The MBCs of BGA against *Streptococcus mutans* strains were 1024 μg/mL for all five strains (Table 1). The MBCs of BGA against *Streptococcus*

*sobrinus* strains 6715 and GTC 278 were 512 and 1024 µg/mL, respectively. The MBC of BGA against other *Streptococcus* strains was 1024 µg/mL, except for *Streptococcus salivarius* HHT (512 µg/mL), *Streptococcus gordonii* DL1 (512 µg/mL), and *Streptococcus oralis* 557 (512 µg/mL). The MBCs of BGA against both *Actinomyces naeslundii* ATCC 12104 and *Actinomyces viscosus* ATCC 15987 were 512 and 1024 µg/mL, respectively. The MBC/MIC ratios for various *Streptococcus* species varied within the range of 2–64. Of the 15 *Streptococcus* species, four had an MBC/MIC ratio of 32, five had an MBC/MIC ratio of 16, and five had an MBC/MIC ratio of 8. The lowest MBC/MIC ratio was 2 for *Streptococcus salivarius* JCM5707 and the highest was 64 for *Streptococcus sobrinus* GTC 278. The MBC/MIC ratios for *Actinomyces* species were 8 and 16 for *A. naeslundii* ATCC 12104 and *A. viscosus* ATCC 15987, respectively. The MBCs of BGA against the Gram-negative rods listed in Table 1 varied from 512 to 2048 µg/mL. The lowest MBC was 512 µg/mL for *Prevotella denticola* JCM8528, whereas the highest was 2048 µg/mL for *Prevotella nigrescens* ATCC 33563. Based on the Clinical and Laboratory Standards Institute guidelines, a drug is considered to exhibit bactericidal activity when the MBC/MIC ratio is ≤ 4, whereas the drug is considered bacteriostatic when the MBC/MIC ratio is ≥ 8 [44,45]. BGA appeared to be bacteriostatic against most oral bacteria but exhibited bactericidal activity against some bacteria: *Streptococcus salivarius* JCM5707, *Aggregatibacter actinomycetemcomitans* JP2, *Prevotella denticola* JCM8528, and *Prevotella intermedia* ATCC 25611.

### BGA reduces the number of viable bacteria in biofilms

Previous studies have demonstrated that the MIC of BGA for supragingival plaques was 128 µg/mL [36]. The same measurements were performed to confirm that the MIC of CPC for supragingival plaque was 40 µg/mL. We established the concentration of BGA at 128 µg/mL and CPC at 40 µg/mL and analyzed their effects on the bacteria in the biofilm. Supragingival plaque bacteria were cultured in six-well polystyrene plates in BHI medium for 24 h to form a biofilm at the bottom of the plate [41]. The biofilm formed was cultured in BHI medium containing BGA, CPC, and DMSO alone as controls (S1 Table) for another 6 h. It was confirmed that 5% DMSO did not reduce viable bacterial count (S1 Fig (A) and (B)).

The number of viable bacteria in the biofilms exposed to DMSO alone implied a CFU of $1.8 \times 10^7$, whereas the number of viable cells in biofilms exposed to CPC was $2.0 \times 10^0$ (below the detection limit of 47.6, P < 0.001, Fig 1). In contrast, the viable bacteria of biofilms exposed to BGA implied a CFU of $4.0 \times 10^5$, which was significantly lower (97.8%) (P < 0.001, Fig 1) than DMSO alone.

These results indicate that BGA acts on the bacteria in biofilms and reduces the number of viable bacteria; however, it is not as effective as CPC.

### Fluorescence microscopy using LIVE/DEAD staining revealed that BGA treatment reduced the number of viable bacteria in biofilms

We analyzed the number of viable and dead bacteria in the biofilms formed by human supragingival plaque bacteria following BGA and CPC application by fluorescence microscopy using LIVE/DEAD staining (Fig 2). The percentage of dead bacteria in the biofilm on the polystyrene plate treated with CPC for 6 h was 85.7% (P < 0.001; Fig 3), whereas the percentage of dead bacteria in the biofilm treated with BGA was 60%. However, BGA demonstrated no significant increase in the percentage of dead bacteria compared with the DMSO group (P = 0.145, Fig 3).

### BGA decreases the amount of biofilm

BGA reduced the number of viable bacteria in biofilms formed by supragingival plaque bacteria (Fig 1). Next, we analyzed the effects of BGA on biofilm formation. The amount of biofilm formed was reduced by 27.6% following BGA application to the BHI medium when compared to that of the control group (P < 0.001; Fig 4). Comparing the amount of reduction in biofilms treated with CPC and BGA, the percentage reduction caused by CPC (31.4%) was nearly the same as that caused by BGA (27.6%) (Fig 4).

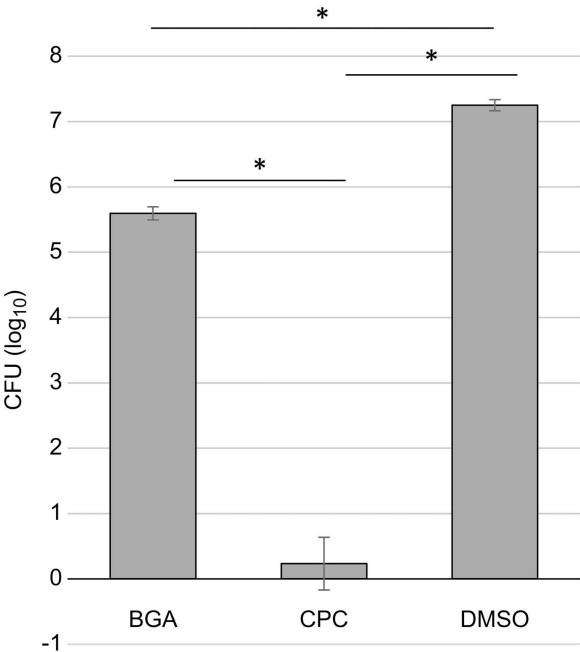

**Fig 1. Colony forming units (CFU) in the biofilms following β-glycyrrhetinic acid (BGA) and cetylpyridinium chloride (CPC) application.**
After 6 h of incubation of the formed biofilm in Brain Heart Infusion medium containing minimum inhibitory concentration (MIC) of BGA or CPC, or only dimethyl sulfoxide as control, the CFUs in the remaining biofilm were measured. These experiments were carried out in triplicate. Error bars denote standard deviation. *P < 0.001.

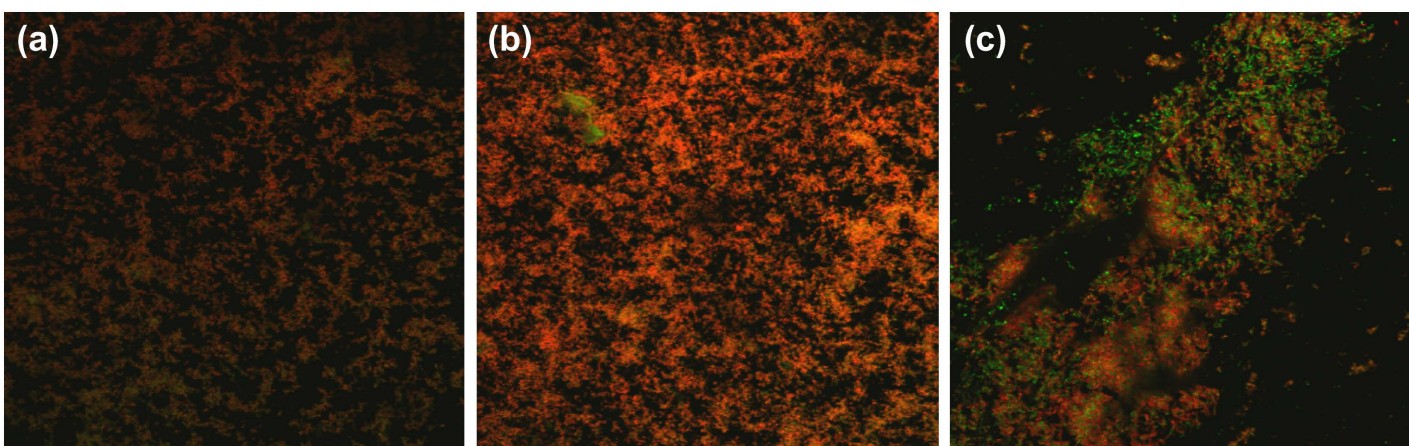

**Fig 2. Representative LIVE/DEAD stained biofilm renderings (x–y plane) following β-glycyrrhetinic acid (BGA) and cetylpyridinium chloride (CPC) application.** Biofilms incubated for 6 h in brain-heart infusion medium containing BGA or CPC at minimum inhibitory concentration (MIC) were LIVE/DEAD stained. Green signal indicates viable live cells (Syto 9) and red signal indicates damaged/dead cells (propidium iodide). All images were taken at 200× magnification. BGA **(A)**, CPC **(B)**, DMSO as negative control **(C)**.

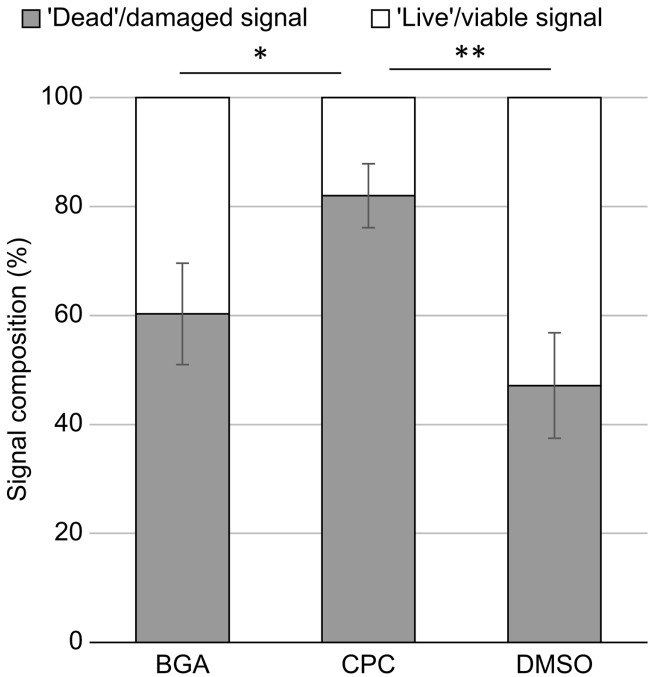

**Fig 3. The ratio of viable and dead bacteria in the biofilms following β-glycyrrhetinic acid (BGA) and cetylpyridinium chloride (CPC) application.** Average percentage signal from biofilms accounted for by dead/damaged (grey bars) and live/viable (white bars) signals in relation to the total signal captured for both. These experiments were carried out in quadruple. Error bars denote standard deviation. *P < 0.05 **P < 0.001.

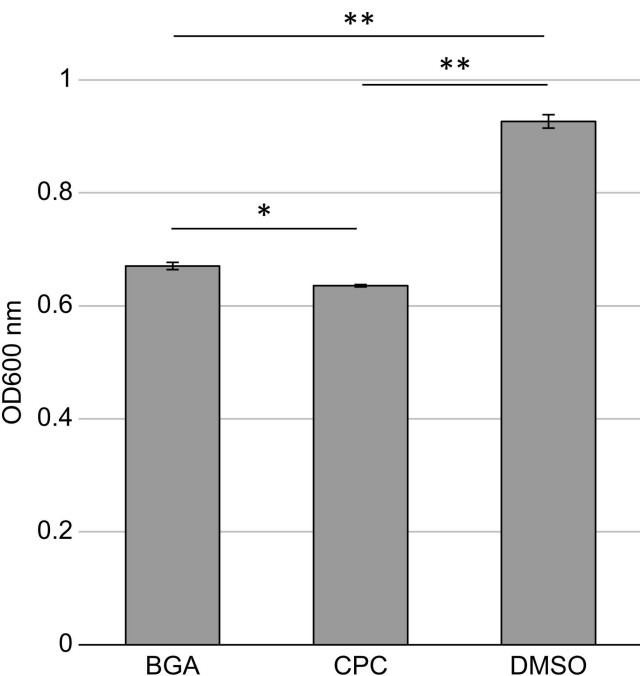

**Fig 4. Amount of biofilm following β-glycyrrhetinic acid (BGA) and cetylpyridinium chloride (CPC) application.** Biofilms incubated for 6 h in brain-heart infusion medium containing minimum inhibitory concentration of BGA, or CPC determined by crystal violet assays. Crystal violet assays were carried out in triplicate. Error bars denote standard deviation. *P < 0.01, ** P < 0.001.

## BGA eliminates biofilms including dead and viable bacteria

Both CPC and BGA reduced biofilm formation (Fig 4), but the differences in the biofilm reduction mechanisms between BGA and CPC are unknown. Therefore, we analyzed the turbidity and number of viable bacteria in the supernatants removed from the biofilms. The turbidity of the culture supernatant of BHI medium containing BGA was the highest ($OD_{600} = 0.237 \pm 0.003$). This was significantly higher than the turbidity following CPC ($OD_{600} = 0.136 \pm 0.002$) and only DMSO ($OD_{600} = 0.096 \pm 0.002$) application ($P < 0.001$, Fig 5(A)). The turbidity of the culture supernatant following CPC application was also higher than that following only DMSO application ($P < 0.001$, Fig 5(A)).

Next, the biofilms were treated with BGA, CPC, or DMSO alone and the number of viable bacteria in culture supernatants was analyzed. Almost no viable bacteria were found in the culture supernatant following CPC treatment, the detection limit was 47.6, the obtained CFUs was below the detection limit. While the culture supernatant following BGA treatment contained viable bacteria with a CFUs of $2.5 \times 10^3$. Viable bacterial counts were significantly higher in the culture supernatant following BGA treatment than following CPC treatment ($P < 0.001$, Fig 5(B)). Although the number of viable bacteria in the BGA group was significantly lower than that in the DMSO group ($P < 0.001$, Fig 5(B)), viable bacteria were still present in the BGA treated samples.

These results indicate that the mechanisms underlying the effects of BGA and CPC on biofilm reduction are different. BGA treatment caused slight turbidity in the supernatant after approximately 2 h (Fig 6(B), S2 Fig), and suspended solids were identified after 4 h (Fig 6(C), S1 Fig). On the other hand, CPC treatment led to suspended solids in the supernatant immediately after addition (Fig 6(A), S2 Fig). Before exposing BGA and CPC to biofilms, the supernatant was removed and washed, suggesting that suspended solids in the supernatant originated from biofilms. Furthermore, the concentrations of BGA and CPC were at respective MIC levels, indicating that bacterial growth in the supernatant was inhibited. Therefore, the turbidity in the supernatant was likely due to detachments from the biofilms, rather than bacterial growth in the supernatant. This suggests that both BGA and CPC can exfoliate biofilms, with BGA exhibiting stronger activity (Fig 5(A)), and with a high proportion of viable bacteria, whereas those exfoliated by CPC treatment were mostly dead (Fig 5(A) and (B)).

## BGA induces physical fragilization of the biofilm

The results indicate that BGA treatment causes biofilm exfoliation via a mechanism different from that of CPC, thus raising the question of whether the physical strength of the biofilm was altered by the BGA or CPC treatment. Therefore, we evaluated the effect of BGA and CPC treatment on the biofilm strength.

The biofilms exposed to BGA, CPC, or only DMSO underwent sonication, and the amount of biofilm remaining at the bottom of the polystyrene plates was measured and compared with biofilms that did not undergo sonication. After sonication, the amount of biofilm in the BGA, CPC, and DMSO groups was significantly lower than that before sonication ($P < 0.001$, S3 Fig (A)). Compared to the DMSO group, the BGA and CPC groups demonstrated significantly less residual biofilm following sonication ($P < 0.01$, $P < 0.001$, respectively, S3 Fig (B)). Furthermore, comparing the BGA and CPC groups, CPC significantly reduced the amount of residual biofilm upon sonication ($P < 0.01$, S3 Fig (B)).

These results indicate that both BGA and CPC treatments induce some degree of fragility of the biofilms to physical impact when compared to DMSO as the control. However, the extent to which fragility to physical impact is involved in the biofilm-exfoliating effect of BGA and CPC treatments is unclear.

## Discussion

In this study, we analyzed the effects of BGA on human supragingival plaque biofilms. BGA primarily demonstrated bacteriostatic action, with bactericidal activity against some bacteria, but weaker bactericidal activity against biofilms than CPC. After BGA exposure, the number of CFUs in the remaining biofilm decreased (Fig 1). However, the ratio of viable to dead bacteria remained unchanged compared to DMSO (Fig 3), which was attributed to the decrease in the amount of biofilm (Fig 4). Furthermore, when the number of viable bacteria in the supernatant was compared to DMSO treated samples,

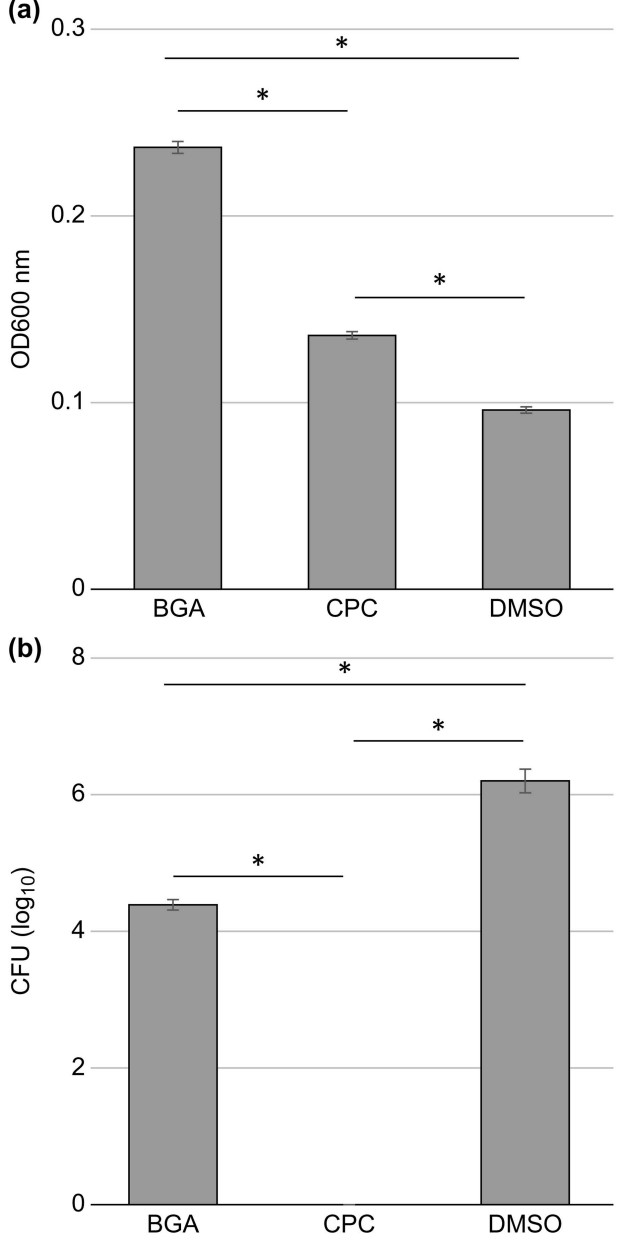

**(a)**

**(b)**

**Fig 5. Suspended solids in supernatant following β-glycyrrhetinic acid (BGA) and cetylpyridinium chloride (CPC) application.** After 6 h of bio-film incubation in brain-heart infusion medium containing minimum inhibitory concentration of BGA or CPC, the absorbance (A) and colony forming units (CFUs) (B) of the supernatant were measured. CFUs following CPC treatment was below the limit of detection. All the experiments were carried out in triplicate. Error bars denote standard deviation. *P < 0.001.

BGA reduced the number of viable bacteria (Fig 5(B)). These results demonstrate the bacteriostatic and bactericidal effects of BGA. However, CPC showed higher bactericidal efficacy than BGA in all cases (Figs 1, 3, 4 and 5(B)). BGA appeared to be bacteriostatic against majority of the oral bacteria (Table 1). These results indicated that BGA is thought to act primarily as a bacteriostatic agent, unlike CPC.

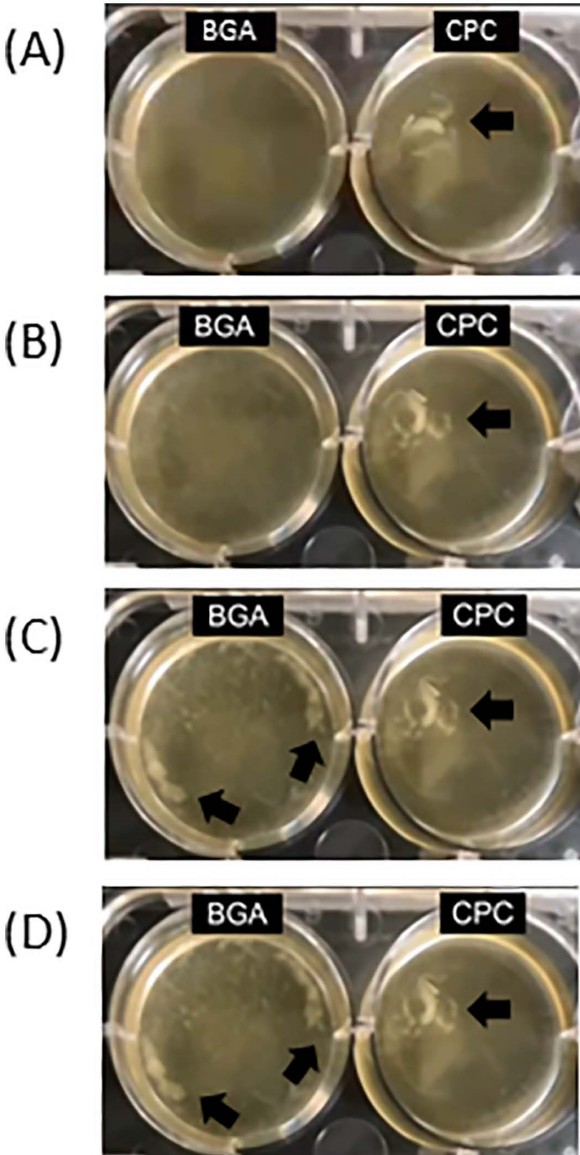

**Fig 6. Changes in the biofilms over time after addition of β-glycyrrhetinic acid (BGA) and cetylpyridinium chloride (CPC).** Time-loop imaging of biofilm changes during incubation in brain-heart infusion medium containing minimum inhibitory concentration of BGA or CPC. Immediately after addition **(A)**, after 2 h **(B)**, after 4 h **(C)**, after 6 h **(D)**.

BGA and CPC significantly reduced the amount of biofilm formed (Fig 4) and BGA demonstrated greater effectiveness than CPC in exfoliating biofilms (Fig 5(A), Fig 6, S2 Fig). In the case of *Vibrio cholerae* biofilms, it has been reported that BGA exposure alters the composition of exopolysaccharides [46]. This may have influenced biofilm exfoliation. However, the present study could not demonstrate chemical or structural changes in the biofilm following BGA exposure. Investigating these aspects would provide more definitive evidence regarding this exfoliation effect. We intend to make the direct visualization of matrix alteration for future research. Furthermore, BGA reportedly affects the biofilm composition and inhibits biofilm formation by acting on quorum sensing [32,46,47]. BGA penetrates the biofilm matrix in *Pseudomonas*

*aeruginosa* and exfoliates biofilms [28]. BGA may have demonstrated this exfoliating effect by acting on quorum sensing in dental plaque, while this has not yet been confirmed. Although different from these mechanisms, enzymes are sometimes used to promote plaque removal [48–50]. Comparing their effects is considered valuable for verifying clinical efficacy and should be pursued in future studies.

In recent years, quorum sensing has received extensive attention as a drug discovery target to find new anti-infection strategies for controlling the spread of resistant bacteria [51,52]. In this study, BGA demonstrated greater effectiveness than CPC in exfoliating biofilms (Fig 5(A, B), Fig 6(A–D), S2 Fig). We hypothesize that BGA likely demonstrates antibiofilm activity by influencing quorum sensing, while CPC exhibits antibiofilm activity through its bactericidal action. Concerns about the emergence of CPC-resistant oral bacteria exist [18], and while no reports of BGA-resistant bacteria have emerged, the possibility cannot be ruled out. The risk of selecting for antimicrobial-resistant bacteria exists with any antimicrobial agent. Nevertheless, oral care products containing BGA are less widely available than those containing CPC. Therefore, we expect BGA to be a new anti-plaque agent, and investigating the relationship between BGA action on quorum sensing and exfoliation of supragingival plaque biofilms is necessary.

This in vitro study to verify the effectiveness of BGA against dental plaque differs from clinical studies in several ways. One key difference was that biofilms derived from supragingival plaque on polystyrene plates were used in this study. This allowed testing with uniform biofilms while still replicating the complex bacterial flora found in the oral cavity. Consequently, BGA and CPC could be compared from various perspectives, including biofilm quantity, viable bacteria count, and bactericidal rate, under identical exposure conditions to the test substances. However, while enzymatic degradation of biofilm was observed on polystyrene plates, some reports indicate no significant difference from placebo formulations in vivo [49,50]. Conversely, Yamashita et al. reported reduction in plaque adhesion after one week of using a toothpaste containing 0.1% BGA [35]. The use of oral compositions containing BGA is expected to inhibit plaque adhesion. In contrast, this study exposed biofilms to a low concentration of 128 µg/ml for 6 hours. This was done to test at the MIC level of BGA, aiming to clarify the mechanism of its effect. We monitored the viable bacteria and biofilm amount over time, and the effect reached a plateau after 6 hours. BGA has also been reported to exhibit bacteriostatic activity and plaque formation inhibition [36]. Therefore, the extent to which the exfoliation effect demonstrated in this study contributes remains unclear. Further verification, such as investigating plaque removal efficacy in vivo, is considered necessary.

This is the first report on the exfoliative effects of BGA on dental plaque. Thus, the use of BGA in oral care products not only as an anti-inflammatory agent but also as an antibacterial agent may be considered. Continuous use of BGA-containing dentifrice for 1 week reportedly resulted in less supragingival plaque and gingival inflammation compared to placebo [35]. However, this is an *in vitro* study, and the exfoliating effect when applied to the oral cavity is unknown. Therefore, investigating the exfoliating effects on dental plaque *in vivo* are necessary. In addition, since the mechanism by which BGA exfoliates biofilms is unknown, we plan to evaluate this effect at the molecular level to elucidate the detailed characteristics of this antimicrobial substance.

In conclusion, BGA demonstrated a primarily bacteriostatic action against supragingival plaque biofilms and superior biofilm-exfoliating effect by a mechanism different from that of CPC. Therefore, BGA is expected to have safe and reliable anti-plaque effects, which may help prevent gingivitis and periodontitis.

## Supporting information

**S1 Table. Test groups and test medium.**
(DOCX)

**S1 Fig. Colony forming units in the biofilms and supernatant following 5.0% dimethyl sulfoxide (DMSO).** The formed biofilm was incubated for 6 h in Brain Heart Infusion medium (BHI) with and without 5.0% DMSO. CFU in (A) the biofilm remaining and (B) supernatant were measured. These experiments were carried out in triplicate. Error bars denote standard deviation. Comparisons with and without DMSO were performed using the t-test.* $P < 0.05$.
(TIF)

**S2 Fig. Time-loop imaging movie of biofilms after addition of β-glycyrrhetinic acid (BGA) and cetylpyridinium chloride (CPC) over time.**
(MP4)

**S3 Fig. The effect of β-glycyrrhetinic acid (BGA) and cetylpyridinium chloride (CPC) treatment on the biofilm strength.** Biofilm biomass following incubation in Brain Heart Infusion medium containing minimum inhibitory concentration of BGA and CPC was determined by crystal violet assays before and after sonication. Absorbance at $OD_{600}$ (A) and ratio of absorbance at $OD_{600}$ before and after sonication (B). All the experiments were carried out in triplicate. Error bars denote standard deviation. *$P < 0.01$, ** $P < 0.001$ (Ultrasound -) †$P < 0.001$ (Ultrasound +) § $P < 0.05$, §§$P < 0.01$, §§§$P < 0.001$. Comparisons before and after ultrasound were performed using t-test. ‡ $P < 0.001$.
(TIF)

## Author contributions

**Conceptualization:** Kayo Sato, Aya Okumura, Nobuo Yoshinari, Akihiro Yoshida.

**Data curation:** Shinya Kato, Xiangtao Ma, Kayo Sato, Aya Okumura.

**Formal analysis:** Shinya Kato, Xiangtao Ma, Kayo Sato.

**Funding acquisition:** Aya Okumura.

**Investigation:** Shinya Kato, Xiangtao Ma.

**Methodology:** Shinya Kato, Kayo Sato, Aya Okumura, Akihiro Yoshida.

**Project administration:** Aya Okumura, Akihiro Yoshida.

**Supervision:** Aya Okumura, Akihiro Yoshida.

**Visualization:** Shinya Kato.

**Writing – original draft:** Kayo Sato, Akihiro Yoshida.

**Writing – review & editing:** Shinya Kato, Xiangtao Ma, Kayo Sato, Aya Okumura, Nobuo Yoshinari, Akihiro Yoshida.

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
