## [Decision Letter · Decision Letter 0]

22 Feb 2026

PONE-D-26-02620Exfoliating effect of β-glycyrrhetinic acid on marginal gingivitis-inducing plaque biofilms: comparison with cetylpyridinium chloridePLOS One

Dear Dr. Yoshida,

Thank you for submitting your manuscript entitled “Exfoliating effect of β-glycyrrhetinic acid on marginal gingivitis-inducing plaque biofilms: comparison with cetylpyridinium chloride” to PLOS ONE.

Your manuscript has now been evaluated by three independent reviewers. Two reviewers consider the study to be of interest and potentially valuable, whereas the third reviewer has raised substantive concerns regarding the adequacy of experimental controls and the clarity of the proposed mechanism of action, and has recommended rejection. After careful consideration of all reports, the Editorial assessment is that the work demonstrates potential merit; however, substantial revisions are required to address the identified methodological and conceptual gaps.

Accordingly, we invite you to submit a Major Revision of your manuscript.

Below is a synthesis of the principal issues that must be addressed in your revised submission and detailed rebuttal.

1. Study Scope and Conceptual Framing

The study characterizes the non-conventional antimicrobial profile of β-glycyrrhetinic acid (BGA) in preformed 24-hour supragingival biofilms. A central contribution of the work is the distinction between the bactericidal activity of cetylpyridinium chloride (CPC) and the apparent physical “exfoliating” or “embrittling” effect attributed to BGA. This conceptual distinction is potentially important for plaque management strategies that do not rely exclusively on direct bacterial lethality.

However, the mechanistic interpretation and terminology require greater precision and consistency throughout the manuscript.

2. Major Methodological Limitations

The reviewers identified several limitations that currently constrain the translational and scientific impact of the study:

• Static versus dynamic biofilm conditions: The use of a static 24-hour biofilm model grown on polystyrene does not adequately reflect the shear forces present in the oral cavity. The limitations of this model must be explicitly acknowledged and discussed.

• Exposure duration: A 6-hour treatment exposure does not reflect clinically relevant mouthwash contact times. The manuscript must address how BGA would be expected to perform under shorter, clinically realistic exposure intervals (e.g., approximately 60 seconds), either experimentally or through a reasoned discussion supported by literature.

• Pooling of plaque samples: Although plaque was collected from 12 volunteers, pooling may obscure inter-individual variability in microbial composition and treatment response. This methodological choice and its implications should be clearly justified and discussed.

3. Critical Scientific Concerns

The following issues must be resolved to meet publication standards:

a. Distinction between “killing” and “removal”

The manuscript reports a reduction in CFUs following BGA treatment while indicating no significant change in the live/dead ratio. This suggests that bacterial cells may be detached from the biofilm surface rather than killed. However, the manuscript at times conflates “disinfection” with “exfoliation.” The terminology must be rigorously standardized throughout to clearly distinguish bactericidal effects from physical removal or structural disruption.

b. Insufficient mechanistic evidence

The claim that BGA induces “embrittlement” of biofilms is not currently supported by direct biochemical or structural evidence demonstrating interaction with the extracellular polymeric substances (EPS) matrix. Without such evidence, the proposed mechanism remains speculative. Additional data or a substantially strengthened mechanistic rationale is required.

4. Experimental Gaps and Required Controls

To address the major concerns raised, the following points must be considered:

• Positive control for biofilm dispersal: The study lacks a comparator known to induce matrix degradation or biofilm dispersal. Inclusion of a positive control (e.g., a matrix-degrading enzyme or a well-characterized surfactant) would allow benchmarking of the exfoliating effect attributed to BGA.

• Negative and vehicle controls: The manuscript must clearly confirm that DMSO concentrations used as vehicle controls do not independently affect biofilm integrity. Inclusion of an additional water or saline control is strongly recommended if not already performed.

• EPS-specific visualization: To substantiate the claim of scaffold disruption, incorporation of EPS-targeted staining (e.g., lectin-based probes) in CLSM analysis is recommended. Direct visualization of matrix alteration would significantly strengthen the mechanistic argument.

5. Revision Guidance

In preparing your revised manuscript, please:

• Reframe the Discussion to emphasize biofilm structural modification rather than general antimicrobial activity, unless bactericidal effects are conclusively demonstrated.

• Address all reviewer comments in a detailed, point-by-point response document.

• Explicitly respond to the concerns that led to the recommendation for rejection and explain how each has been resolved.

We look forward to receiving your revised manuscript.

Kind regards,

Abdelwahab Omri, Pharm B, Ph.D

Academic Editor

PLOS One

Journal Requirements:

“I have read the journal's policy and the authors of this manuscript have the following competing interests: Kayo Sato and Aya Okumura are paid employees of Kao Corporation, Tokyo, Japan. This research was funded by Kao Corporation. The authors declare no conflicts of interest.”

We note that one or more of the authors are employed by a commercial company: Kao Corporation

Reviewers' comments:

Reviewer's Responses to Questions

**Comments to the Author**

1. Is the manuscript technically sound, and do the data support the conclusions?

Reviewer #1: Partly

Reviewer #2: Partly

Reviewer #3: Partly

2. Has the statistical analysis been performed appropriately and rigorously? 

Reviewer #1: Yes

Reviewer #2: I Don't Know

Reviewer #3: Yes

3. Have the authors made all data underlying the findings in their manuscript fully available?

Reviewer #1: Yes

Reviewer #2: Yes

Reviewer #3: Yes

4. Is the manuscript presented in an intelligible fashion and written in standard English?

Reviewer #1: Yes

Reviewer #2: Yes

Reviewer #3: No

5. Review Comments to the Author

Reviewer #1: This manuscript shows the effects of β-glycyrrhetinic acid (BGA) on supragingival plaque biofilms and compares these effects with those of cetylpyridinium chloride (CPC). The use of complex plaque biofilms is meaningful for clinical application. However, the conclusion that BGA shows a “superior exfoliating effect” appears to be somewhat overstated based on presented results. After appropriate revisions in response to the points, the suitability for publication should be reconsidered.

1. The authors conclude that BGA showes a stronger biofilm-exfoliating effect than CPC. However, an increase in turbidity may reflect not only detached biofilms but also cell debris or aggregates of dead bacteria. Therefore, I think that a greater turbidity does not necessarily indicate a superior exfoliating effect. The authors should clearly describe these limitations and discuss the results more cautiously, avoiding overinterpretation.

2. Iimportant finding of this study is that BGA treatment induces detachment of biofilm components containing viable bacteria. However, in vivo setting, such bacteria may retain the potential for re-adhesion to make biofilms. Therefore, biofilm detachment alone may not reflect a clinical benefit. In addition, the potential application of BGA in oral care products may require more cautious discussion. This point should be addressed in the Discussion.

3. The concentrations of BGA (128 µg/mL) and CPC (40 µg/mL) were determined based on their respective MIC values, which is scientifically reasonable. However, it remains unclear whether these concentrations are clinically relevant and whether comparisons at the same MIC appropriately reflect differences in biofilm-exfoliating activity. The Discussion should clarify the rationale and limitations of the concentration settings and the interpretation of MIC-based comparisons.

4. The discussion of quorum sensing is based on previous studies and is not directly supported by the present results. Some statements may imply quorum sensing involvement and should be more clearly presented as speculation.

5. In Fig. 6, arrows or brief annotations indicating detached biofilm aggregates would improve clarity.

6. In Fig. 5B, the detection limit of the CFU assay should be clearly indicated.

Reviewer #2: The following corrections are required

1. Lines 224–226: Revise all claims of “bactericidal action” through out manuscript because MIC/MBC ratios and LIVE/DEAD data show BGA is mainly bacteriostatic, not bactericidal.

2. Lines 227: Correct the misreporting of bacterial strains (e.g., JP2 is not Prevotella denticola according to Table 1) 3. Line 232: The author has declared that all the data is available but here mentioned -"Data not shown". Please justify not including the data, update the claim or provide the data if possible and relevant for the article.

4. Line 394-395: Tone down translational claims (e.g., prevention of periodontal disease) because in vitro biofilm data alone do not justify clinical implications.

5. Line199-201: Clarify whether assumptions for parametric tests ANOVA and t-test (normality, homogeneity) were tested.

6. A few typos and grammatical errors need corrections, for ex- Line 232: "date not shown"...here its date or data, Line 245: viable bacterial, Line 288: "biofilms biomass" should be replaced with 'biofilm biomass', Line 306: CFU unmeasurable should be replaced with -'CFU were below the limit of detection' etc.

Reviewer #3: This study investigated the in vitro effects of BGA on preformed supragingival plaque biofilms and compared its efficacy that with of CPC. The authors reported that BGA reduced viable bacteria and exhibited superior biofilm exfoliation compared to CPC, suggesting a distinct mechanism of action. However, the reviewer has several concerns regarding this manuscript.

1. The title contains ‘marginal gingivitis’. What classification was used for ‘marginal gingivitis’?

2. Introduction: In P3L43-44, there is a statement “The disruption of this homeostatic host-bacteria relationship occurs during gingivitis, marking the initiation of periodontitis [6,7].”. However, is there any evidence in Refs. 6 and 7 to support the idea that the progression of gingivitis causes periodontitis?

3. Introduction: The concern regarding the ‘emergence of CPC-resistant oral bacteria’ is emphasized repeatedly. Is there no concern regarding the emergence of BGA-resistant oral bacteria?

4. Materials and methods: Supragingival plaque was collected from 12 healthy participants. Why were samples not collected from patients with gingivitis?

5. Materials and methods: Minimum bactericidal concentrations (MBCs) were determined using the microdilution method, in which BGA was adjusted to 0, 8, 16, 32, 64, 128, 256, 512, and 1024 μg/mL. How was the MBC of 2048 μg/mL measured in Table 1?

6. Materials and methods: The amount of exfoliated biofilm was determined by measuring the absorbance of the supernatants at OD600 after incubation in the test media for 6 h. However, OD600 was affected by bacterial growth in the supernatants. How do the authors justify the method to consider the absorbance of the supernatants at OD600 as the amount of exfoliated biofilm?

7. Materials and methods: How were the 24h/6h incubation periods determined?

8. Materials and methods: P14L227, ‘Prevotella denticola JP2’ or ‘Aggregatibacter actinomycetemcomitans JP2’?

9. Materials and methods: P14L228, ‘Prevotella denticola ATCC25611’ or ‘Prevotella intermedia ATCC25611’?

10. Results: P15L242, What does “which was significantly (97.8 %) lower (P<0.001, Fig. 1) than that of BGA exposed to DMSO alone.” mean? Was it a comparison with the BGA or DMSO group?

11. Results: P18L294-295, It was stated that “both CPC and BGA reduced biofilm formation to the same degree”; however, Fig 4 shows that there is a significant difference between these two groups.

12. Discussion: P22L368-369: There is a statement that “BGA demonstrated greater effectiveness than CPC in exfoliating biofilms.”. However, OD600 should be affected by the bactericidal and bacteriostatic properties of the test media. How were these bactericidal and bacteriostatic properties excluded from turbidity?

13. The discussion acknowledges the in vitro nature of the study and the unknown exfoliating effects in vivo. However, it could be more comprehensive by discussing other limitations, such as the specific concentrations used, fixed incubation times, or generalizability of results from polystyrene plates to complex oral surfaces.

14. English editing is required throughout the manuscript. The following are examples:

P6L99. ‘Aggraegatibacter’ should be ‘Aggregatibacter’.

P8L120. ‘Flanklin’ should be ‘Franklin’.

P14L232. ‘(date not shown)’ should be ‘data not shown)’.

6. PLOS authors have the option to publish the peer review history of their article (what does this mean?). If published, this will include your full peer review and any attached files.

Reviewer #1: No

Reviewer #2: No

Reviewer #3: No

---

## [Author Response · Author response to Decision Letter 1]

14 Apr 2026

EMID: beb8b9fc8494d88d

Reply to the Editor

Thank you for your important comments and suggestions.

We have addressed them as follows.

1. Study Scope and Conceptual Framing

The mechanistic interpretation and terminology require greater precision and consistency throughout the manuscript.

We revised the manuscript to clarify the differences between BGA and CPC (line 361-362 and 370-371).

2. Major Methodological Limitations

The limitations of these three models (Static vs. dynamic biofilm conditions; exposure duration; pooling of plaque samples) must be discussed.

This in vitro study to verify the effectiveness of BGA against dental plaque differs from clinical studies in several ways. One key difference was that biofilms derived from supragingival plaque on polystyrene plates were used in this study. This allowed testing with uniform biofilms while still replicating the complex bacterial flora found in the oral cavity. Consequently, BGA and CPC could be compared from various perspectives, including biofilm quantity, viable bacteria count, and bactericidal rate, under identical exposure conditions to the test substances. However, while enzymatic degradation of biofilm was observed on polystyrene plates, some reports indicate no significant difference from placebo formulations in vivo. Conversely, Yamashita et al. reported reduction in plaque adhesion after one week of using a toothpaste containing 0.1% BGA. The use of oral compositions containing BGA is expected to inhibit plaque adhesion. In contrast, this study exposed biofilms to a low concentration of 128 μg/ml for 6 hours. This was done to test at the MIC level of BGA, aiming to clarify the mechanism of its effect. We monitored the viable bacteria and biofilm amount over time, and the effect reached a plateau after 6 hours. BGA has also been reported to exhibit bacteriostatic activity and plaque formation inhibition. Therefore, the extent to which the exfoliation effect demonstrated in this study contributes remains unclear. Further verification, such as investigating plaque removal efficacy in vivo, is considered necessary.

This description has been added to the “DISCUSSION” (line 399-417).

Another reason for pooling of plaque sample was that it enabled consistent experimentation with a uniform biofilm, requiring minimal sample collection per individual.

This description has been added to “MATERIALS AND METHODS” (line 119-120).

3. Critical Scientific Concerns

a. Distinction between "killing" and "removal" is unclear.

We standardized terminology throughout the manuscript.

b. Mechanistic evidence is insufficient.

In the case of Vibrio cholerae biofilms, it has been reported that BGA exposure alters the composition of exopolysaccharides. This may have influenced biofilm exfoliation. However, the present study could not demonstrate chemical or structural changes in the biofilm following BGA exposure. Investigating these aspects would provide more definitive evidence regarding this exfoliation effect.

We added this limitation and the need for further study to the “DISCUSSION” (line 374-398).

4. Experimental Gaps and Required Controls

To benchmark the exfoliating effect of BGA, include a positive control.

Although different from these mechanisms, enzymes are sometimes used to promote plaque removal. Comparing their effects is considered valuable for verifying clinical efficacy and should be pursued in future studies.

This statement was added to “DISCUSSION” (line 383-386).

The manuscript must clearly confirm that DMSO concentrations used as vehicle controls do not independently affect biofilm integrity.

We confirmed that 5% DMSO did not reduce viable bacterial counts (S1 Fig. (A) and (B)).

This text has been added to the “RESULTS” (lines 238-239).

To substantiate the claim of scaffold disruption, incorporation of EPS-targeted staining in CLSM analysis is recommended

We intend to make the direct visualization of matrix alteration for future research.

This description has been added to “DISCUSSION” (line 378-379).

5. Revision Guidance

Reframe the Discussion to emphasize biofilm structural modification rather than general antimicrobial activity, unless bactericidal effects are conclusively demonstrated.

We revised the manuscript to clarify the differences between BGA and CPC (line 361-362, 368, 370-371, and 390-392).

Address all reviewer comments in a detailed, point-by-point response document, and explicitly respond to the concerns that led to the recommendation for rejection.

This Response to Reviewers addresses all comments and describes how they were resolved.

Reply to Reviewer 1

Thank you for your important comments and suggestions. We have addressed them as follows.

1. Does the degree of turbidity reflect the biofilm exfoliation effect?

Before exposing BGA and CPC to biofilms, the supernatant was removed and washed, suggesting that suspended solids in the supernatant originated from biofilms. Furthermore, the concentrations of BGA and CPC were at MIC levels, indicating that bacterial growth in the supernatant was inhibited. Therefore, we interpret turbidity as reflecting biofilm detachment.

This text has been added to the “RESULTS” (line 317-322).

2. Please discuss the potential application of BGA in oral care products. Is the biofilm detachment the only clinically relevant factor?

Yamashita et al. reported that reduction in plaque adhesion after one week of using a toothpaste containing 0.1% BGA. BGA has been reported to exhibit bacteriostatic activity and plaque formation inhibition. Therefore, the extent to which the biofilm exfoliating effect demonstrated in this study contributes remains unclear. Further verification, such as investigating plaque removal efficacy in vivo, is considered necessary.

This text has been added to the “DISCUSSION” (lines 407-417).

3. Please discuss the rationale and limitations of the concentration settings and the interpretation of MIC-based comparisons.

Yamashita et al. reported that reduction in plaque adhesion after one week of using a toothpaste containing 0.1% BGA. In contrast, we exposed biofilms to BGA at 128 ug/mL for 6 hours to test BGA at its MIC level and to help clarify its mechanism.

This text has been added to the “DISCUSSION” (lines 407-412).

4. Clarify that the discussion of quorum sensing is speculative.

We added statements indicating that quorum sensing as a mechanism for the BGA detachment effect is a hypothesis and has not been confirmed (lines 382-383 and lines 390-391).

5. Add arrows or brief annotations indicating detached biofilm aggregates in Fig. 6.

We added arrows indicating detached biofilm aggregates in Fig. 6.

6. Add the detection limit of the CFU assay in Fig. 5B.

We added that the detection limit of the CFU assay is 47.6 to the “RESULTS” (line 242 and lines 306-307).

Reply to Reviewer 2

Thank you for your important comments and suggestions. We have addressed them as follows.

1. Revise all claims of "bactericidal action" throughout the manuscript.

We revised the manuscript to state that the primary effect of BGA is bacteriostatic (line 226, line 361, line 368, lines 370-371 and line 427).

2. Correct the misreporting of bacterial strains.

We have corrected the misreporting of bacterial strains (line 74, line 77, lines 99-100, line 218, and lines 227-229).

3. Resolve the contradiction involving the phrase "data not shown."

Because the MIC of CPC for supragingival plaque was 40 ug/mL, we deleted "data not shown" (line 233).

4. Tone down translational claims (lines 394-395).

Given that dental plaque accumulation contributes to periodontal disease, we think that BGA helps prevent gingivitis and periodontitis by reducing plaque levels. We revised the statement to be more cautious regarding clinical implications (lines 429-430).

5. Clarify whether assumptions for parametric tests were checked.

Normality was tested by Shapiro–Wilk test.

This was added to “MATERIALS AND METHODS” (line2 202-203).

6. Correct typos and grammatical errors.

We corrected typos and grammatical errors (line 226, line 307, line 361, line 368, lines 370-371 and line 427).

Reply to Reviewer 3

Thank you for your important comments and suggestions. We have addressed them as follows.

1. Title: Revise the classification of gingivitis.

In accordance with the American Academy of Periodontology and the European Federation of Periodontology classification, we replaced "marginal gingivitis" with "plaque-induced gingivitis" (line 1, line 4, line 44, line 45, and line 51). We also added Ref. 8 as supporting evidence (lines 455-457).

2. Introduction: Include that gingivitis progression can lead to periodontitis.

We added Ref. 8, which states that gingivitis is a precursor of periodontitis (line 44, and lines 455-457).

3. Introduction: Is there concern regarding the emergence of BGA-resistant oral bacteria?

Concerns about the emergence of CPC-resistant oral bacteria exist, and while no reports of BGA- resistant bacteria have emerged, the possibility cannot be ruled out. Few anti-biofilm agents target quorum sensing, and they are currently being researched as a new approach against drug-resistant bacteria. Therefore, we expect BGA to be a new anti-plaque agent.

This text has been added to the “DISCUSSION” (lines 392-396).

4. Materials and Methods: Why were samples not collected from patients with gingivitis?

To investigate the efficacy in preventing gingivitis and periodontitis, bacterial communities adjacent to the gingival margin were collected from participants who did not have gingivitis or periodontitis.

This text has been added to “MATERIALS AND METHODS” (lines 112-114).

5. Materials and Methods: How was the MBC of 2048 ug/mL measured in Table 1?

When the MBC was 1024 ug/mL or higher, the BGA concentration was adjusted to 0, 256, 512, 1024, 2048, and 4096 μg/ml, and the experiment was repeated using the same procedure.

This text has been added to Materials and Methods (lines 144-146).

6. Materials and Methods: Justify using OD600 of supernatants as the amount of detached biofilm.

Before exposing BGA and CPC to biofilms, the supernatant was removed and washed, suggesting that suspended solids in the supernatant originated from biofilms. Furthermore, the concentrations of BGA and CPC were at MIC levels, indicating that bacterial growth in the supernatant was inhibited. Therefore, the turbidity in the supernatant was likely due to detachments from the biofilms, rather than bacterial growth in the supernatant.

This text has been added to the “RESULTS” (lines 317-322).

7. Materials and Methods: How were the 24 h/6 h incubation periods determined?

Supragingival plaque was incubated for 24 hours to form biofilms, consistent with a previous study. Reference was added at line 151, line 159, line 175, line 236 and lines 569-571.

For BGA exposure, we monitored viable bacteria and biofilm amount over time, and the effect plateaued after 6 hours; therefore, we set the exposure time to 6 hours.

This text has been added to the “DISCUSSION” (lines 412-413).

8. Materials and Methods: "Prevotella denticola JP2" or "Aggregatibacter actinomycetemcomitans JP2"?

We revised this to Aggregatibacter actinomycetemcomitans JP2 (line 228).

9. Materials and Methods: "Prevotella denticola ATCC25611" or "Prevotella intermedia ATCC25611"?

We revised this to Prevotella intermedia ATCC25611 (line 229).

10. Results: Was it a comparison with the BGA group or the DMSO group (P15L242)?

This was a comparison with DMSO alone. We revised the sentence to: "which was significantly (97.8%) lower (P<0.001, Fig. 1) than DMSO alone." (lines 242-244).

11. Results: Revise the sentence "both CPC and BGA reduced biofilm formation to the same degree" (P18L294-295).

We revised the sentence to: "Both CPC and BGA reduced biofilm formation". (line 296).

12: Discussion: OD600 could be affected by bactericidal and bacteriostatic properties. How were these excluded?

Before exposing BGA and CPC to biofilms, the supernatant was removed and washed, suggesting that suspended solids in the supernatant originated from biofilms. Furthermore, the concentrations of BGA and CPC were at MIC levels, indicating that bacterial growth in the supernatant was inhibited. Therefore, the turbidity in the supernatant was likely due to detachments from the biofilms, rather than bacterial growth in the supernatant.

This text has been added to the RESULTS (lines 317-322).

13. Discussion: Discuss other limitations (s the specific concentrations used, fixed incubation times, or generalizability of results from polystyrene plates to complex oral surfaces).

This is in vitro study to verify the effectiveness of BGA against dental plaque, differs from clinical studies in several ways. One key difference was that biofilms derived from supragingival plaque on polystyrene plates were used in this study. This allowed testing with uniform biofilms while still replicating the complex bacterial flora found in the oral cavity. Consequently, BGA and CPC could be compared from various perspectives, —including biofilm quantity, viable bacteria count, and bactericidal rate, —under identical exposure conditions to the test substances. However, while enzymatic degradation of biofilm was observed on polystyrene plates, some reports indicate no significant difference from placebo formulations in vivo. Conversely, Yamashita et al. reported reduction in plaque adhesion after one week of using a toothpaste containing 0.1% BGA. The use of oral compositions containing BGA is expected to inhibit plaque adhesion. In contrast, this study exposed biofilms to a low concentration of 128 μg/ml for 6 hours. This was done to test at the MIC level of BGA, aiming to clarify the mechanism of its effect. We monitored the viable bacteria and biofilm amount over time, and the effect reached a plateau after 6 hours. BGA has also been reported to exhibit bacteriostatic activity and plaque formation inhibition. Therefore, the extent to which the exfoliation effect demonstrated in this study contributes remains unclear. Further verification, such as investigating plaque removal efficacy in vivo, is considered necessary.

This description has been added to “DISCCUTION” (line 399-417).

14. Comment: English editing is required throughout the manuscript.

We obtained professional English editing and re-checked the manuscript to correct remaining errors. We also corrected specific terms (e.g., “Pseudomonas aeruginosa” at line 74, “Klebsiella pneumoniae” at line 77, “Aggregatibacter actinomycetemcomitans” at lines 99-100, and “Streptococcus salivarius JCM5707” at line 218) and removed "Frankrin" and "data not shown," as appropriate.

---

## [Editor Report · Decision Letter 1]

16 Apr 2026

Exfoliating effect of β-glycyrrhetinic acid on plaque inducing gingivitis: comparison with cetylpyridinium chloride

PONE-D-26-02620R1

Dear Dr. Akihiro Yoshida,

We’re pleased to inform you that your manuscript has been judged scientifically suitable for publication and will be formally accepted for publication once it meets all outstanding technical requirements.

Kind regards,

Abdelwahab Omri, Pharm B, Ph.D

Academic Editor

PLOS One

Additional Editor Comments (optional):

Accept
---

## [Editor Report · Acceptance letter]

PONE-D-26-02620R1

PLOS One

Dear Dr. Yoshida,

I'm pleased to inform you that your manuscript has been deemed suitable for publication in PLOS One. Congratulations! Your manuscript is now being handed over to our production team.

Kind regards,

on behalf of

Dr. Abdelwahab Omri

Academic Editor

PLOS One